# Pseudosparse neural coding in the visual system of primates

Sidney R. Lehky [1,2 ✉], Keiji Tanaka[1] & Anne B. Sereno [3,4]

When measuring sparseness in neural populations as an indicator of efficient coding, an implicit assumption is that each stimulus activates a different random set of neurons. In other words, population responses to different stimuli are, on average, uncorrelated. Here we examine neurophysiological data from four lobes of macaque monkey cortex, including V1, V2, MT, anterior inferotemporal cortex, lateral intraparietal cortex, the frontal eye fields, and perirhinal cortex, to determine how correlated population responses are. We call the mean correlation the *pseudosparseness index*, because high pseudosparseness can mimic statistical properties of sparseness without being authentically sparse. In every data set we find high levels of pseudosparseness ranging from 0.59–0.98, substantially greater than the value of 0.00 for authentic sparseness. This was true for synthetic and natural stimuli, as well as for single-electrode and multielectrode data. A model indicates that a key variable producing high pseudosparseness is the standard deviation of spontaneous activity across the population. Consistently high values of pseudosparseness in the data demand reconsideration of the sparse coding literature as well as consideration of the degree to which authentic sparseness provides a useful framework for understanding neural coding in the cortex.

[1] Cognitive Brain Mapping Laboratory, RIKEN Center for Brain Science, Wako-shi, Saitama 351-0198, Japan. [2] Computational Neurobiology Laboratory, The Salk Institute, La Jolla, CA 92037, USA. [3] Department of Psychological Sciences, Purdue University, West Lafayette, IN 47907, USA. [4] Weldon School of Biomedical Engineering, Purdue University, West Lafayette, IN 47907, USA. ✉email: sidney@salk.edu

The efficient coding hypothesis of Barlow[1], in which a neural code is optimized to minimized the number of spikes needed to transmit a given signal, postulates high sparseness in populations of visual neurons. Efficient coding under criteria based on Shannon information theory has become an important concept organizing thinking about visual processing[2–6]. Neurophysiological studies have characterized sparseness and other measures of efficient coding across various areas of the visual cortex[7–18], as well as the lateral geniculate nucleus[19] and retina[20]. In addition to vision, the concepts of efficient coding and sparseness have been applied to data from a variety of other domains, including audition[21], olfaction[22], somatosensation[23], and memory[24].

Sparseness is a function of the probability distribution of neural responses. Responses with high sparseness have responses that are disproportionately at the tails of the probability distribution (heavy-tailed) relative to a Gaussian distribution. Neural responses with low sparseness have a distribution with thin tails. A variety of sparseness measures exist in the literature[9,13,14,25]. The sparseness measure used in this study is the reduced kurtosis of the response distribution, which has previously been used in a number of theoretical and experimental studies[14,26–28]:

$$\text{Sparseness} = \frac{\sum\limits_{i=1}^{N}(r_i - \bar{r})^4}{(N-1)s^4} - 3. \quad (1)$$

Here, $r_i$ is the response of the $i$th cell in the population to a single stimulus and $N$ is the number of cells in the population. Mean response across the population to a single stimulus is $\bar{r}$, and the standard deviation of the responses is indicated by $s$. The reduced kurtosis is the regular statistical kurtosis with three subtracted so that the Gaussian distribution is normalized to have a value of 0. Sparseness >0 indicates increasingly sparse coding.

Population sparseness is sparseness for a single stimulus across a neural population (with average sparseness calculated over the stimulus set). This differs from lifetime sparseness, sometimes called neural selectivity[14], which is determined by the probability distribution of a single neuron to a set of stimuli. We are concerned here with population sparseness and not lifetime sparseness.

The expectation during sparse coding is that different neurons in the population are activated by different stimuli. In other words, during sparse coding the response vectors for a given neural population should ideally be uncorrelated for different stimuli. This is illustrated by Fig. 1a, where responses to the same model neural population are shown to different stimuli. Each circle in a horizontal row represents a neuron in the population, and gray levels indicate responses (activities) to a stimulus. Vertically, the responses for different stimuli for the neural population are shown. Under authentic sparseness, population response vectors for different stimuli are uncorrelated, so that a different random set of neurons is activated for each stimulus.

Population response statistics are shown in the form of a *population response spectrum* in Fig. 1b. The *x*-axis gives index numbers for members of the population, not necessarily in any particular order. The *y*-axis gives the mean (black line) and standard deviation (gray shading) of responses for each neuron in the population over all members of the stimulus set. Because response vectors for different stimuli during authentic sparseness are uncorrelated, the population spectrum is flat, and standard deviations are large.

Next, we show the median probability density function (pdf) of neural responses across the model population for single stimuli during authentic sparseness (Fig. 1c). Each stimulus produces a single pdf across the neural population, and we show the median

pdf from all the members of the stimulus set. We are looking at pdfs across a neural population and not lifetime pdfs of a single neuron across a stimulus set.

The significance of the pdf for our purposes is that the pdf defines population sparseness. There are a variety of sparseness measures, as mentioned earlier, but in essence they boil down to some measure on the shape of the population pdf. If different conditions (both receptive field properties and the stimulus set) lead to the same population response pdf, then they have the same sparseness.

Having described the situation where the population response is uncorrelated for different stimuli (Fig. 1, left column), we present the opposite extreme where the population response to different stimuli is perfectly correlated (Fig. 1, right column; why this correlated situation is important will be made clear below). In contrast to the uncorrelated population response spectrum (Fig. 1b), which is flat, the correlated population response spectrum (Fig. 1e) is bumpy reflecting different mean responses for different members of the population. Also, because population responses are completely correlated for different stimuli, the population response spectrum (Fig. 1e) has no variance. The median pdf of responses for all the members of the correlated stimulus set is shown in Fig. 1f.

The important point to emphasize here is that response pdfs can be essentially identical for uncorrelated stimuli (Fig. 1c) and for perfectly correlated stimuli (Fig. 1f). And as the pdfs determine sparseness, uncorrelated and correlated responses can generate the same levels of sparseness. Although both cases show equal measures of sparseness, only the uncorrelated case (Fig. 1, left column) corresponds to genuine sparseness. The correlated case exhibits what we call *pseudosparseness*.

We define pseudosparseness as the mean correlation coefficient for population response vectors over the stimulus set. For example, given a population size of 100 neurons, the population response for a single stimulus will be a vector of length 100. Each stimulus will generate a different response vector, and the mean correlation of all pairs off response vectors for the stimulus set can be calculated to give the pseudosparseness. If the response vectors are uncorrelated, pseudosparseness will be 0.0 (Fig. 1b). If the response vectors are completely correlated, pseudosparseness will be 1.0 (Fig. 1e).

Sparseness values are identical under the authentic sparseness condition (Fig. 1b) and under the pseudosparseness condition (Fig. 1e) in this example, being equal to 1.705 (as measured by the reduced kurtosis, Eq. (1), of the response pdfs in Fig. 1c, f). Yet despite having identical sparseness, the population responses produce radically different values of pseudosparseness.

Although sparseness levels have been measured in different areas of the visual system, as described above, there has been no attempt to distinguish between authentic sparseness and pseudosparseness. Here, we examine the issue by reanalyzing single-electrode and multielectrode neurophysiological data for various areas in four different lobes of monkey cortex (occipital, temporal, parietal, and frontal) responsive to synthetic and natural visual stimuli, including V1, V2, MT, anterior inferotemporal cortex (AIT), lateral intraparietal cortex (LIP), the frontal eye field (FEF), and perirhinal cortex (Prh).

## Results

**Pseudosparseness in the monkey visual cortex**. We examined 12 neurophysiological data sets (15 data sets total when including all subsets of data) collected from various regions of visually responsive cortex in macaque monkeys, all of them stimulated with various visual patterns. For each data set, we plotted the population response spectrum and calculated the

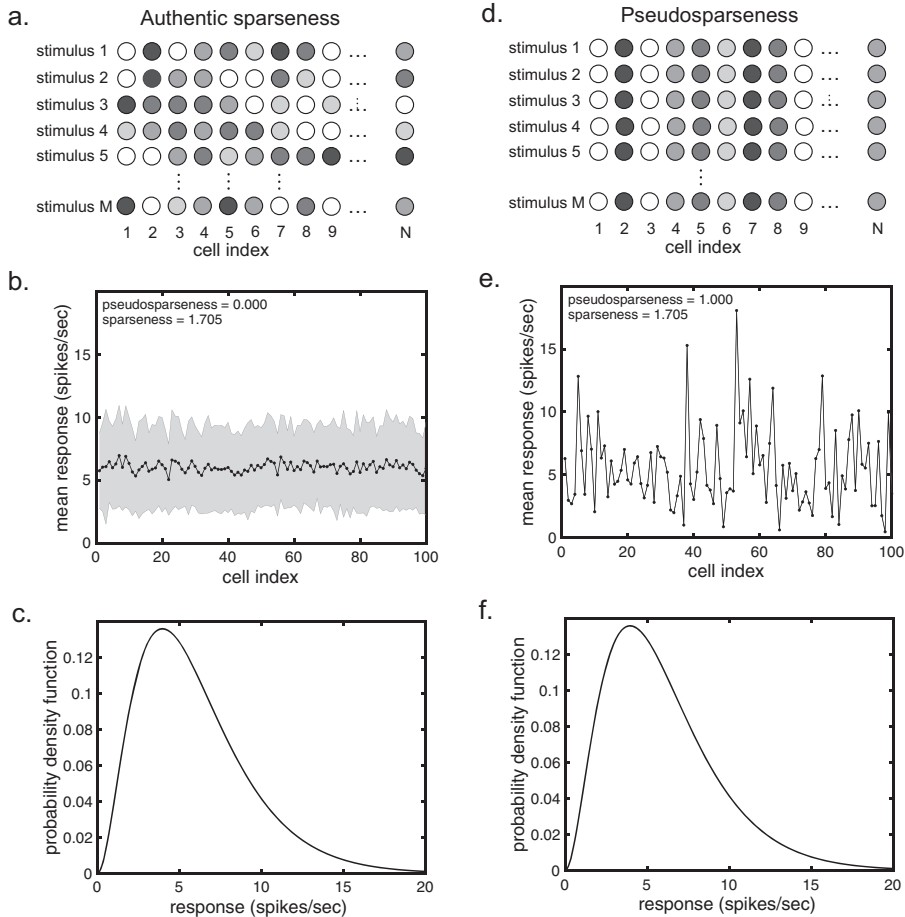

**Fig. 1 Comparison between sparseness and pseudosparseness. a–c** Example showing authentic sparseness. **a** Responses of the same neural population to different stimuli, under authentic sparseness. Each row of circles shows the responses to a different stimulus, with gray levels indicating response levels. Population responses to different stimuli are uncorrelated and a different random set of neurons is activated by each stimulus. **b** An example population response spectrum for authentic sparseness. Mean stimulus response (black) and response standard deviation (gray shading) for each neuron in the population is plotted on the y-axis, with neurons assigned arbitrary index numbers along the x-axis. For this example, each stimulus produces a set of responses across the population described by a gamma distribution. Parameters for the gamma distribution are identical for each member of the stimulus set. The pseudosparseness index calculates the mean correlation coefficient between population response vectors between all members of the stimulus set. **c** Median probability density function (pdf) for the neural population responses for individual stimuli under authentic sparseness. Each stimulus produces a single pdf for the responses across the neural population, and then the median pdf from all stimuli is determined and shown. **d–f** Example showing pseudosparseness. **d** Responses of the same neural population to different stimuli, under pseudosparseness. Under pseudosparseness, population response vectors for different stimuli are correlated, so that the same set of neurons is always activated for all stimuli. **e** An example population response spectrum for pseudosparseness. The population spectrum is produced using the same gamma distribution for population responses as used for authentic sparseness. For pseudosparseness, however, population response vectors for different stimuli are perfectly correlated. Therefore, the population spectrum has a standard deviation of zero, and pseudosparseness = 1.0. **f** Median probability density function for individual stimulus responses under maximum pseudosparseness. Note that this probability density function is identical to that under authentic sparseness in **c**. In the plots for the model $N = 100$ neurons with $n = 10,000$ stimuli per neuron.

pseudosparseness and sparseness indices. In all cases, the pseudosparseness index was high, in excess of 0.59 (on a scale 0.0–1.0). This indicates that cortical neural population response vectors are highly correlated for different stimuli, contrary to the general assumption when interpreting population sparseness measures that population responses for different stimuli are uncorrelated (i.e., have a pseudosparseness index = 0.0). Among the 15 response spectra for the different data sets, the Pearson correlation between sparseness and pseudosparseness was $r = 0.441$, significantly different from zero ($p = 0.016$ based on bootstrap resampling, 95% confidence interval (CI) = [0.124, 0.734], power > 0.99). The correlation coefficient was likely reduced because it pooled data from different experimental designs, and involved different population sizes.

Population response spectra from V1 using multielectrode data[29,30] are shown from one monkey, using grating stimuli (pseudosparseness = 0.782 ± 0.003, Fig. 2a) and natural stimuli (pseudosparseness = 0.778 ± 0.004, Fig. 2b), both with the same set of neurons ($N = 66$) for the two stimulus conditions. The response spectrum for a second monkey ($N = 44$) is shown for grating stimuli (pseudosparseness = 0.594 ± 0.008, Fig. 2c) and natural stimuli (pseudosparseness = 0.669 ± 0.006, Fig. 2d). Pooled data for the two monkeys are shown for grating stimuli (pseudosparseness = 0.722 ± 0.004, Fig. 2e) and natural stimuli (pseudosparseness = 0.721 ± 0.004, Fig. 2f). Pseudosparseness is essentially the same for gratings and natural stimuli. (Pseudosparseness values are given as mean ± SD, where standard deviation (SD) is determined by bootstrap resampling).

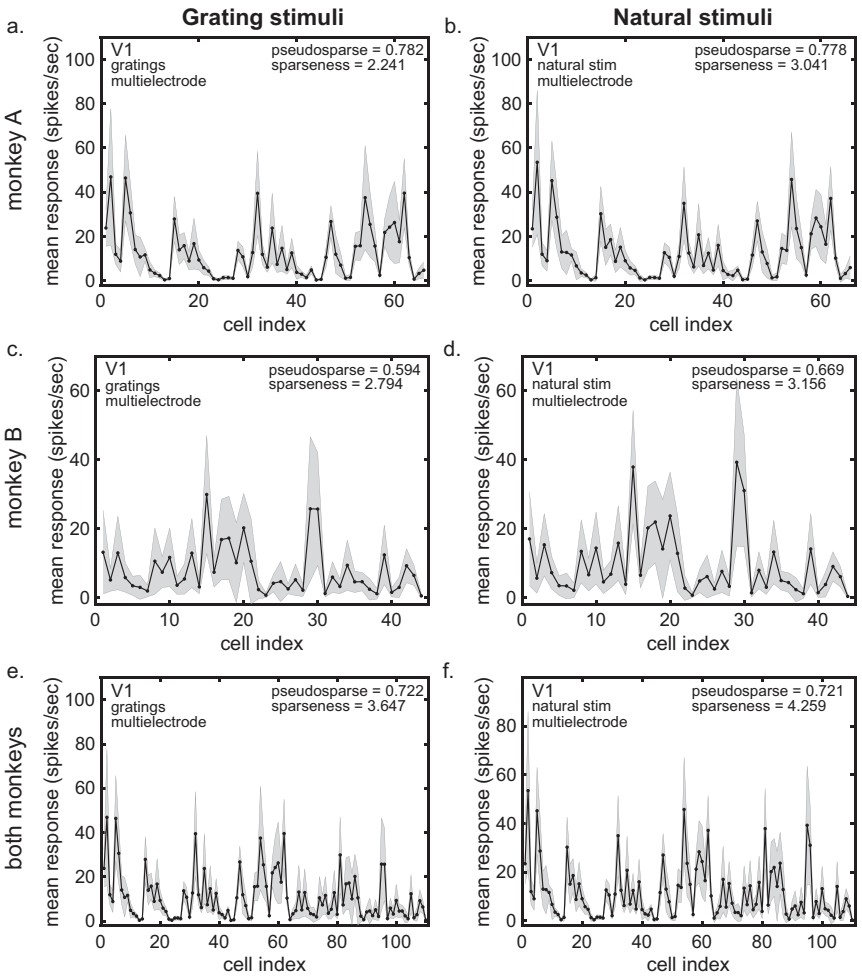

**Fig. 2 Response spectra for V1, using multielectrodes with grating and natural stimuli. a** Monkey A response spectrum using grating stimuli. $N = 66$ neurons with $n = 416$ stimuli per neuron. **b** Monkey A response spectrum using natural stimuli. $N = 66$ neurons with $n = 540$ stimuli per neuron. **c** Monkey B response spectrum using grating stimuli. $N = 44$ neurons with $n = 416$ stimuli per neuron. **d** Monkey B response spectrum using natural stimuli. $N = 44$ neurons with $n = 540$ stimuli per neuron. **e** Pooled data from both monkeys using grating stimuli. **f** Pooled data from both monkeys using natural stimuli. Neurons were the same for each monkey when using grating or natural stimuli (Data from Coen-Cagli et al.[29]). Mean pseudosparseness and median sparseness values (median because sparseness values are strongly right skewed) are given here and in subsequent figures. Gray error bars represent standard deviation.

Going beyond the pseudosparseness index, the entire response spectra themselves, using grating stimuli (Fig. 2a) and natural stimuli (Fig. 2b), are almost identical. Again, these two response spectra involve an identical population of neurons, but using different stimulus sets. The Pearson correlation between the response spectra (calculated neuron by neuron for the respective spectra) is 0.987 (95% CI = [0.979, 0.992]). Response spectra for the second monkey are shown in Fig. 2c, d, with correlation 0.989 (95% CI = [0.979, 0.994]).

It is possible that population statistics, such as pseudosparseness, may depend on whether the population is recorded simultaneously (using multielectrodes) or sequentially (using single electrodes). In order to compare multielectrode data with single-electrode data, we took ten multielectrode recording sessions (involving three monkeys) all with identical stimuli, selected one neuron randomly from each session and synthesized a sequential population of $N = 10$ neurons. Single-electrode populations were repeated 10,000 times using a different random sample of neurons from the various sessions. Example single-electrode populations are shown for grating stimuli (Fig. 3a) and natural stimuli (Fig. 3b). Average population statistics over the 10,000 replications were pseudosparseness = $0.715 \pm 0.113$

(gratings) and $0.727 \pm 0.115$ (natural images). We also have a single-electrode V1 sample from a different data set[14,31] involving a single monkey using synthetic stimuli (Fig. 3c), with a similar value of pseudosparseness = $0.742 \pm 0.009$.

Pseudosparseness values as indicated above were comparable for multielectrode (Fig. 2a, b) and single-electrode data (Fig. 3a, b) in the same data set. In contrast, sparseness values were much larger for multielectrode data than for single-electrode data. For multi-electrodes, sparseness = $2.241 \pm 2.254$ (SE = 0.111) for gratings, and sparseness = $3.041 \pm 2.792$ (SE = 0.120) for natural images. For single-electrodes, sparseness = $-0.721 \pm 0.628$ (SE = 0.029) for gratings, and sparseness = $-0.587 \pm 0.665$ (SE = 0.031) for natural images.

Since population size for the multielectrode data (Fig. 2a, b) was $N = 66$ and population size for the single-electrode data (Fig. 3a, b) was $N = 10$, we examined if population size affected population statistics. This was done by randomly subsampling the neurons from the multielectrode population into populations of different sizes, repeating each subpopulation 10,000 times.

Pseudosparseness as a function of population size for multi-electrode data are plotted for gratings (Fig. 4a) and natural stimuli (Fig. 4b). Different populations sizes were created by randomly

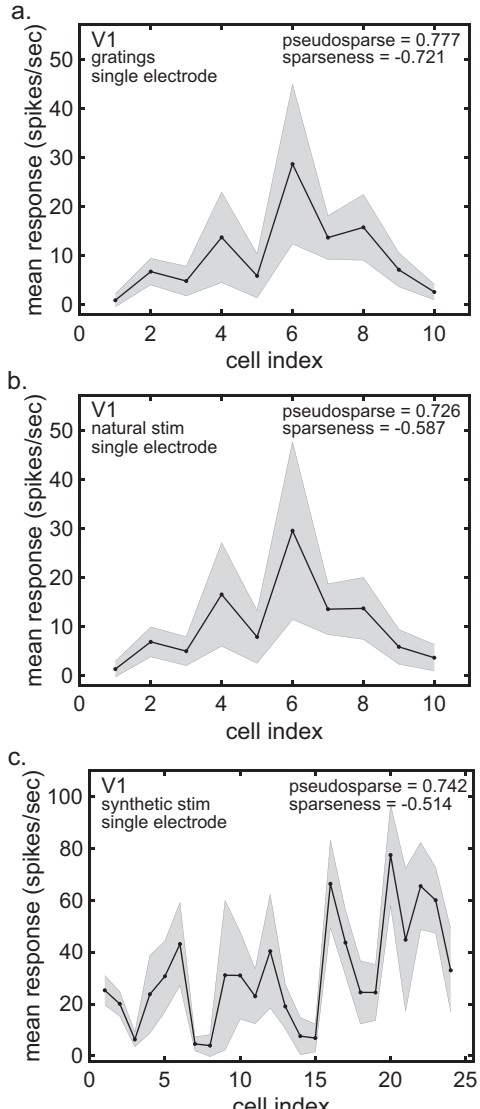

**Fig. 3 Response spectra for V1 using single-electrode data. a** Response spectrum using grating stimuli. $N = 10$ neurons with $n = 416$ stimuli per neuron. **b** Response spectrum using natural stimuli. $N = 10$ neurons with $n = 540$ stimuli per neuron. Neurons in **a** and **b** are the same. These single-electrode data were synthesized from multielectrode data (Data from Coen-Cagli et al.[29]). **c** Response spectrum using synthetic stimuli. $N = 24$ neurons with 157 stimuli per neuron (Data from Lehky et al.[14,31]). Gray error bars represent standard deviation.

subsampling the data from Fig. 2a, b. When matching the single-electrode population size ($N = 10$), the single-electrode pseudosparseness is about one standard deviation lower than the multielectrode value. The difference (multi–single) has 95% CI $= [-0.057, 0.251]$ for gratings and CI $= [-0.083, 0.235]$ for natural stimuli, based on bootstrap resampling. The difference is not statistically significant against a null hypothesis of equal means ($p = 0.110$ for gratings, $p = 0.144$ for natural stimuli, two-sided) based on bootstrap resampling, power > 0.99 in both cases. Thus, the electrode configuration (single vs. multi) is not a major factor affecting pseudosparseness (i.e., average population correlation) measurements.

Sparseness as a function of population size is plotted for gratings (Fig. 4c) and natural stimuli (Fig. 4d). Sparseness is more sensitive to population size than pseudosparseness because

sparseness depends on the tails of the neural response probability density function across a population, and small populations do not allow accurate estimations of the tails. For sparseness, the multielectrode and single-electrode data are in very close agreement given matching population sizes. The difference (multi–single) has 95% CI $= [-0.745, 1.300]$ for gratings and CI $= [-0.705, 1.518]$ for natural stimuli, based on bootstrap resampling. The difference is not statistically significant against a null hypothesis of equal means ($p = 0.353$, power > 0.99 for gratings; $p = 0.615$, power $= 0.835$ for natural stimuli, two-sided). Again, electrode configuration is not a major factor affecting sparseness measurements.

In addition to V1, we have pseudosparseness values obtained from a variety of extrastriate areas, including V2 (refs. [32–34]; pseudosparseness $= 0.700 \pm 0.064$, Fig. 5a), MT[35,36] (pseudosparseness $= 0.621 \pm 0.010$, Fig. 5b), Prh cortex[37] (pseudosparseness $= 0.813 \pm 0.006$, Fig. 5c), and area TE in AIT cortex[37] (pseudosparseness $= 0.642 \pm 0.012$, Fig. 5d). The AIT and Prh data were collected with the same set of object stimuli. For all these extrastriate areas pseudosparseness is very high, far greater than zero, as was the case for V1. Note that the MT data mimic free viewing by introducing simulated saccades in a video using natural stimuli (similar to the sparse coding study by Vinje and Gallant[10]). The MT response spectrum using these natural stimuli does not appear to be notably different from the other data.

We have a second population response spectrum using shape stimuli from AIT[12] (Fig. 6a), with a pseudosparseness index of $0.815 \pm 0.098$, similar to the value produced by the first AIT data set described above. In addition to data from the ventral visual stream in AIT, we show the population response spectrum of data collected from the dorsal stream in LIP[12,38], using the same set of shape stimuli (Fig. 6b). These LIP data produced a pseudosparseness index of $0.978 \pm 0.011$. Finally, we show the population response spectrum of FEF data with again the same shape stimuli[39] (Fig. 6c), with a pseudosparseness index of $0.971 \pm 013$.

In addition to examining the pseudosparseness index using shape stimuli in different cortical areas, we also looked at responses to the retinotopic location of stimuli. The population response spectrum for stimulus location in AIT data[11,40] is shown in Fig. 6d, with a pseudosparseness index of $0.938 \pm 0.030$. For LIP data[11,38], the location response spectrum is shown in Fig. 6e, with a pseudosparseness index of $0.593 \pm 0.144$. Finally, for FEF (previously unpublished data from Peng et al.[39]), the location response spectrum is shown in Fig. 6f, with a pseudosparseness index of $0.943 \pm 0.025$.

**Pseudosparseness model.** The purpose of this model is to reproduce typical values of the pseudosparseness index observed in the cortical data. The model consists of a population of two-dimensional Gaussian receptive fields activated by a set of stimuli. Each stimulus produces a population response vector. The mean correlation coefficient is calculated to provide the pseudosparseness index for the given stimulus set and receptive field set. For our purposes, the nature of the feature space defined by the model is arbitrary. The 2D receptive fields may be, for example, in physical space (denoting stimulus locations), shape space, color space, or motion space, and so forth.

The receptive field population forms a hexagonal grid of overlapping cells. Each Gaussian receptive field response is defined by:

$$r = G(\mu_G, \sigma_G)e^{-\left(\frac{(x-x_0)^2 + (y-y_0)^2}{2\sigma^2}\right)} + O(\mu_O, \sigma_O), \quad (2)$$

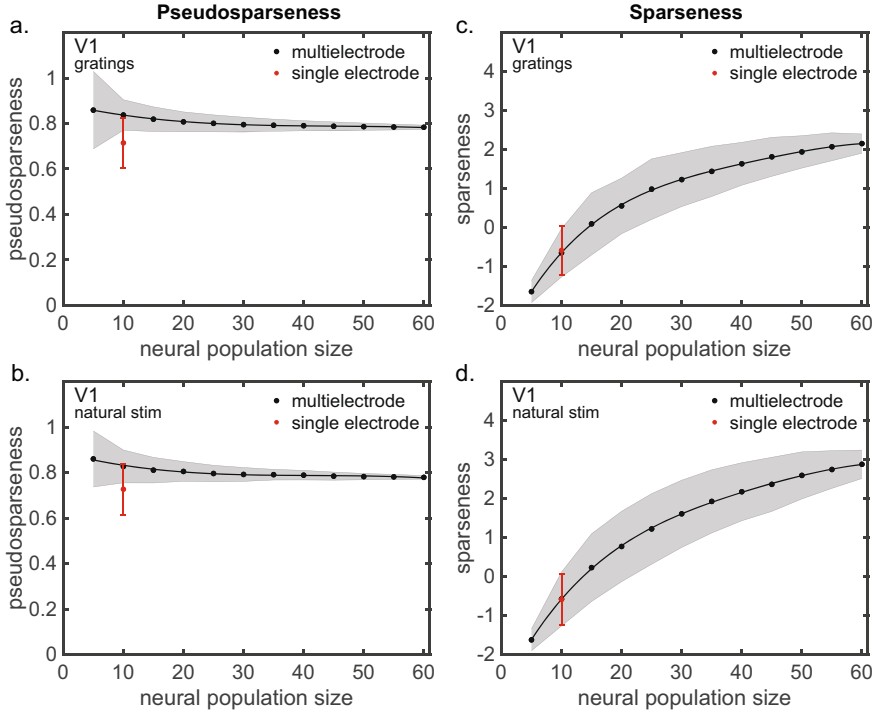

**Fig. 4 Comparison between multielectrode and single-electrode V1 data for pseudosparseness and sparseness values as a function of population size.**
**a** Comparison of pseudosparseness using grating stimuli. **b** Comparison of pseudosparseness using natural stimuli. **c** Comparison of sparseness using grating stimuli. **d** Comparison of sparseness using natural stimuli. Single-electrode population was synthesized from multielectrode data by selecting one cell from different multielectrode recording sessions from the same data set. Multielectrode data was subsampled to produce various population sizes. Entire curves were generated for completeness, but the critical comparison is that between the $N = 10$ single-electrode population and the $N = 10$ multielectrode population. $N = 5$–$60$ neurons on the x-axis with 10,000 resampled population activities based on $n = 416$ stimuli (gratings) or $n = 540$ stimuli (natural images) for each neuron (Data from Coen-Cagli et al.[29]).

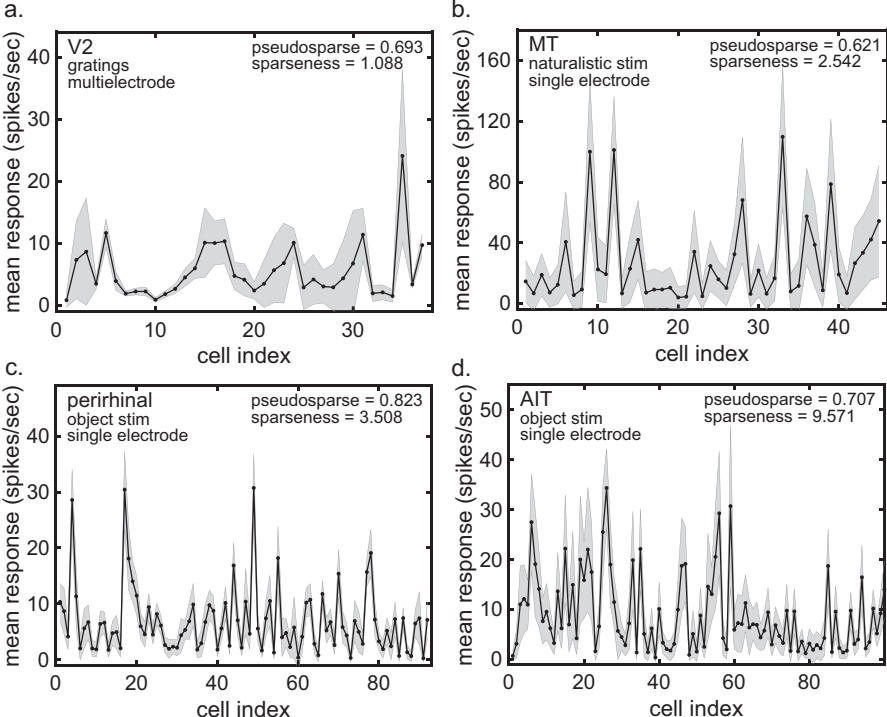

**Fig. 5 Response spectra for four extrastriate cortical areas using a variety of stimuli.** **a** Response spectrum from V2. $N = 37$ neurons with 8 stimuli per neuron (Data from Zandvakili and Kohn[32] and Semedo et al.[33]). **b** Response spectrum from MT. $N = 45$ neurons with 200 stimuli per neuron (Data from Nishimoto and Gallant[36]). **c** Response spectrum from perirhinal cortex. $N = 92$ neurons with $n = 110$ stimuli per neuron (Data from Lehky and Tanaka[37]). **d** Response spectrum from anterior inferotemporal cortex (AIT). $N = 122$ neurons with $n = 110$ stimuli per neuron (Data from Lehky and Tanaka[37]). Gray error bars represent standard deviation.

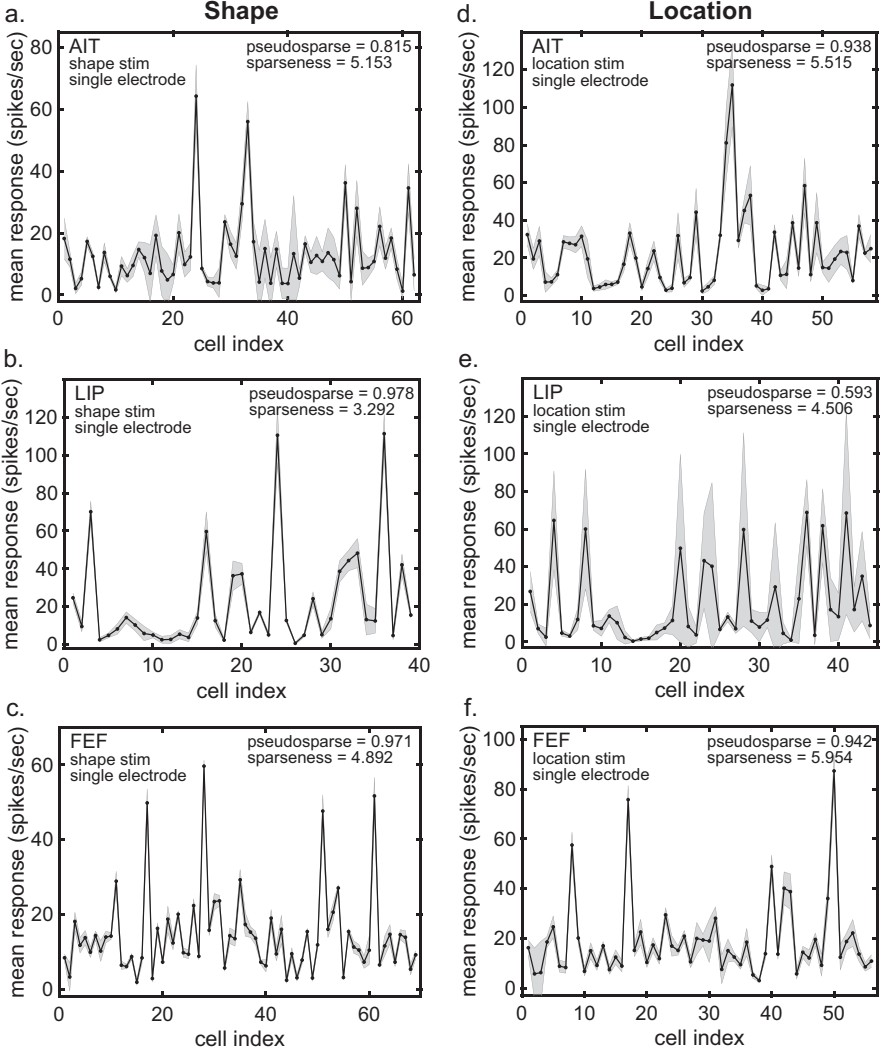

**Fig. 6 Response spectra for three cortical areas using either shape or retinotopic location stimuli. a** Anterior inferotemporal cortex (AIT) data for stimulus shape. $N = 85$ neurons with $n = 8$ stimuli per neuron (Data from Lehky and Sereno[12]). **b** Lateral intraparietal cortex (LIP) data for stimulus shape. $N = 53$ neurons with $n = 8$ stimuli per neuron (Data from Lehky and Sereno[12]). **c** Frontal eye field (FEF) data for stimulus shape. $N = 72$ neurons with $n = 8$ stimuli per neuron (Data from Peng et al.[39]). **d** AIT data for stimulus location. $N = 83$ neurons with $n = 8$ stimuli per neuron (Data from Lehky et al.[40] and Sereno and Lehky[11]). **e** LIP data for stimulus location. $N = 65$ neurons with $n = 8$ stimuli per neuron (Data from Lehky et al.[40] and Sereno and Lehky[11]). **f** FEF data for stimulus location $N = 62$ neurons with $n = 8$ stimuli per neuron (Previously unpublished data). Gray error bars represent standard deviation.

where $(x_0, y_0)$ is the receptive field center in feature space. The receptive field radius is given by $\sigma$, indicating one space constant of the receptive field. The response for each neuron has a multiplicative gain $G$ and an additive offset $O$. Offset is the spontaneous activity or baseline activity for each neuron. Heterogeneity in responses across the population is introduced by randomly setting gain and offset for each neuron in a Gaussian fashion, so that $G(\mu_G, \sigma_G)$ has mean gain $\mu_G$ and standard deviation of the gain $\sigma_G$, while $O(\mu_O, \sigma_O)$ has mean offset $\mu_O$ and standard deviation of the offset $\sigma_O$.

Two other parameters define the receptive field population. The first is receptive field spacing, the distance between receptive field centers (in feature space, not necessarily in physical space), denoted by $\eta$. The second is *receptive field dispersion*, how spread out receptive field centers are across the feature space (in essence the diversity of receptive fields in feature space). In retinotopic space, for example, dispersion would indicate how far receptive fields centers are spread out from the fovea[11,41]. For other feature spaces (spatial frequency, orientation, etc.), dispersion measures the range

of receptive field centers within that feature space. The receptive field dispersion is a diameter in feature space denoted by $\gamma$.

In addition to the properties of the receptive field population, the model defines properties of the stimulus set. Each stimulus is a point in the feature space (we don't need to specify it any more than that for present purposes). A region of feature space containing the entire stimulus set is the *stimulus set field*. The number of stimuli in the stimulus set is $n$, and the $n$ members of the stimuli are uniformly distributed over the stimulus field. For convenience we define the stimulus set field to be a region of feature space enclosed by a circle. The diameter of this circle is the *stimulus set dispersion*, denoted by $\phi$. The size of the stimulus dispersion indicates the amount of diversity within the stimulus set.

Each stimulus activates a point on the receptor mosaic and activates the overlapping Gaussian receptive fields to varying degrees, forming a population response vector for that stimulus. The set of population response vectors for all stimuli allows plotting of the population response spectrum for the model, and calculation of the pseudosparseness and sparseness values.

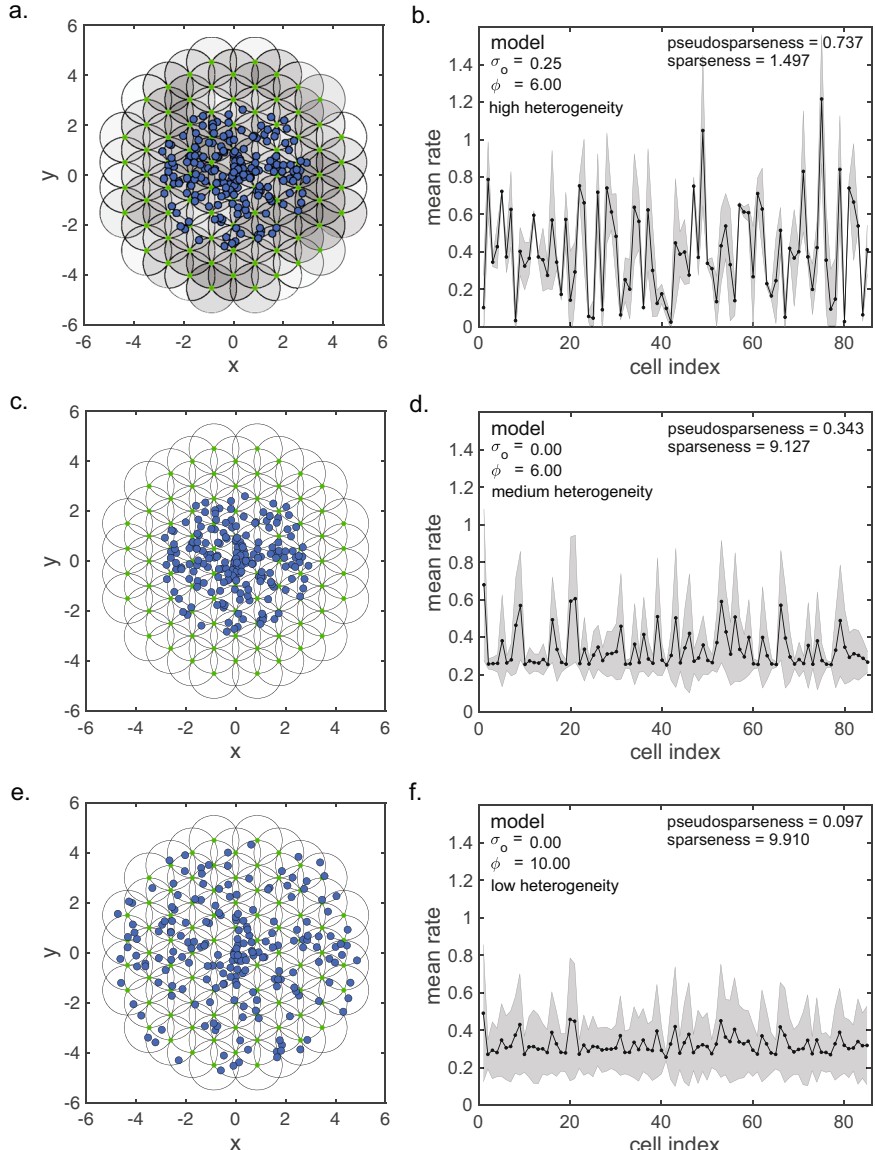

**Fig. 7 Model for generating various level of pseudosparseness in a population of receptive fields activated by random stimuli.** Top row: high pseudosparseness conditions. **a** Hexagonal grid of overlapping receptive fields in a 2D feature space (e.g., position, shape, etc.). Each receptive field has a Gaussian tuning curve in the feature space described in Eq. (2), with a small green dot indicating receptive field center and circle indicating receptive field drawn at space constant $\sigma = 2$ (in arbitrary units). Receptive spacing is $\eta = 1.0$ and receptive field dispersion (size of circular boundary of green dots) is $\gamma = 10.0$. Additional parameter values are gain mean $\mu_G = 1.0$, gain standard deviation $\sigma_G = 0.25$, offset mean $\mu_O = 0.25$, and offset standard deviation $\sigma_O = 0.25$. Blue dots indicate a random set of stimuli. Stimulus set dispersion (size of circular boundary of blue dots) is $\phi = 6$. **b** Resulting response spectrum, showing high level of pseudosparseness = 0.737. Second and third rows: examples of two conditions producing lower pseudosparseness, namely, reducing the response offset level and increasing the stimulus set dispersion. These conditions make population responses less heterogeneous. **c** Receptive field mosaic with offset standard deviation reduced to $\sigma_O = 0.0$, indicated by receptive field gray levels being white. **d** Resulting response spectrum, showing a lower pseudosparseness compared to when response offset level is higher. **e** Receptive field mosaic with stimulus set dispersion increased to $\phi = 10$, in addition to response offset again set to zero. **f** Increasing the stimulus set dispersion leads to further decrease in pseudosparseness. Gray error bars represent standard deviation.

An example configuration of the model is shown in Fig. 7a. The black circles show Gaussian receptive fields (drawn at the level of $\sigma$ in Eq. (2)), while the blue dots indicate stimuli. For this example the RF diameter was $\sigma = 2$ and the spacing between RF centers was $\eta = 1$ (both in arbitrary units), while the number of stimuli was $n = 200$. The full set of model parameters are listed in the caption for Fig. 7a. The population response spectrum resulting from this model is shown in Fig. 7b and has high pseudosparseness = 0.737.

Pseudosparseness is sensitive to parameters that change the statistical heterogeneity of neural response across a population, while not being sensitive to parameters that do not change heterogeneity. Pseudosparseness is sensitive to the response gain $G(\mu_G, \sigma_G)$ and offset level $O(\mu_O, \sigma_O)$, the dispersion of receptive fields $\gamma$ (the diversity of receptive fields tunings in the feature space), and the dispersion of the stimulus set $\phi$ (the diversity of the stimulus set in the feature space). On the other hand, pseudosparseness is not sensitive to the number of stimuli in the

stimulus set $n$, receptive field diameter $\sigma$, and receptive field spacing $\eta$.

Among parameters for receptive field response, the key variable is the standard deviation $\sigma_O$ of the response offsets (i.e., standard deviation of spontaneous activities or baseline activities) across the population. Increasing $\sigma_O$ leads to an increase in pseudosparseness. On the other hand, changing the mean $\mu_O$ of the offsets across the population does not make a large difference. Regardless of whether mean offset is high or low, the pseudosparseness is about the same.

With respect to response gain across the receptive population, the mean gain $\mu_G$ is important as it modulates the effect of the offset downward, so that increasing $\mu_G$ causes the pseudosparseness to decrease. Increasing $\mu_G$ decreases the effectiveness of the offset standard deviation $\sigma_O$ through the relationship $\sigma_O/\mu_G$ (the coefficient of variation of $\sigma_O$ relative to $\mu_G$). Increasing $\mu_G$ produces a higher base of activity, against which offsets are relatively smaller. On the other hand, the standard deviation of the gain $\sigma_G$ is not a major variable.

As an example, decreasing the offset standard deviation (going from $\sigma_O = 0.25$ to $\sigma_O = 0.0$, holding all else constant) lowers pseudosparseness from 0.737 to 0.328. This is seen upon comparing the response spectrum in Fig. 7b to d. Decreasing standard deviations of offsets (baseline activities) leads to decreased heterogeneity across the neural encoding population, reducing the tendency of neurons to have mean responses over all stimuli that are systematically higher or systematically lower than for other neurons. This produces lower pseudosparseness.

Pseudosparseness also decreases as stimulus set dispersion becomes greater relative to the receptive field dispersion. Stimulus set dispersion is the diversity and richness contained within the stimulus set, the breadth of the feature space being covered. When stimulus set dispersion is relatively small ($\phi = 6$, Fig. 7c), the circle enclosing the stimulus set (large blue dots) is smaller than the circle enclosing the receptive field centers (small green dots). Under this situation, receptive fields beyond the stimulus set field are inadequately stimulated, leading to increased heterogeneity in the responses across the neural population and consequent high pseudosparseness. When stimulus set dispersion is increased to better match the receptive field dispersion ($\phi = 10$, Fig. 7e), there is less heterogeneity across the neural population responses, leading to lower pseudosparseness (Fig. 7f). As an alternative to increased stimulus set dispersion, decreased receptive field dispersion relative to stimulus set dispersion in a cortical area could also lead to decreased pseudosparseness.

Possibly an effect of relatively small stimulus set dispersion is seen in the shape data in Fig. 6 (AIT, LIP, and FEF, left column). These data had a small stimulus set (eight stimuli) leading to small stimulus dispersions, and predictions of higher pseudosparseness values under this model.

In summary, large statistical heterogeneity across population responses in the model (i.e., high response offsets, small stimulus dispersion, and high receptive field dispersion, Fig. 7a) produces a population exhibiting high pseudosparseness under well-defined conditions. Such high pseudosparseness is comparable to the high pseudosparseness demonstrated in the neurophysiological data.

## Discussion

Previous literature in visual neurophysiology reports high sparseness/efficient coding in nonhuman primate cortical recordings. We show that pseudosparseness, or correlation between population responses to different stimuli, can also result in measures of artifactually high sparseness. We show that there are consistently high levels of pseudosparseness across visually responsive cortical regions from striate to frontal cortex, using different stimulus classes (e.g., synthetic vs. natural stimuli, or shape vs. location), as well as for single-electrode and multielectrode data. Such high pseudosparseness suggests what has been reported as high sparseness in the previous literature may in some cases be high pseudosparseness. These findings necessitate a closer examination and reconsideration of the prior literature on sparse coding, as well as reflection on to what extent authentic sparseness provides a useful framework for understanding neural coding in cortex. There are limitations and trade-offs to sparse coding as an organizational principle for neural coding[42] allowing room for alternatives to be considered, including those not based on Shannon information theory, such as algorithmic information (Kolmogorov complexity)[43].

When characterizing population statistics (such as sparseness and pseudosparseness), a persistent concern in the field has been the effect of using single-electrode vs. multielectrode recording techniques. When a population is synthetized sequentially over several single-electrode recording sessions, it is possible that changes in general arousal or attention during different sessions leads to inhomogeneities in the population compared to when the population is recorded in parallel using multielectrodes. However, upon examining this issue, we found virtually no difference between sparseness recorded with single-electrodes compared to multielectrodes and a small decrease (one standard deviation) in pseudosparseness recorded with single-electrodes.

Our finding that response characteristics remain consistent across sessions are compatible with observations that properties of identified individual neurons in monkey inferotemporal cortex remain stable, when chronically recorded for weeks or months[44]. We believe that the effect of using single-electrodes vs. multielectrodes depends on the interneuronal correlations involved in the data analysis. When no correlations are involved, which is a first-order analysis of the population (as in sparseness), single-electrode and multielectrode data match. When correlations between responses occur in the analysis (a second-order or higher-order analysis of the population), then single electrodes should show lower values of a population variable relative to multielectrodes. If the analysis involves correlations with respect to mean responses across a population (as opposed to trial-to-trial correlations), then the effects of using single electrodes may be modest, as is the case for pseudosparseness. If trial-to-trial correlations across populations are needed for analysis, then multielectrode techniques are obviously essential because in such cases comparisons must be carried out simultaneously.

Using the same V1 neural population, we show that the response spectra to synthetic stimuli (e.g., gratings) and natural stimuli are almost identical (Fig. 2). Response spectra were examined not only by measuring the pseudosparseness values for each spectrum, but also by calculating the correlation coefficient between population responses for the two stimulus sets. The close similarity of responses raises the possibility that pseudosparseness, in this case, may reflect some low-level biophysical characteristic across the population. In particular, response offset (essentially spontaneous or baseline activity), which is an important factor in the pseudosparseness model, may be a fixed biophysical characteristic for individual neurons in the population (although variable for different neurons across the population), thus leading to similar profiles for different stimulus sets.

High levels of cortical pseudosparseness can mimic sparseness. We distinguish here between classical, authentic sparseness in neural population representations (Fig. 1, left column) and what we call pseudosparseness (Fig. 1, right column), which can mimic certain statistical properties of authentic sparseness. With authentic sparseness, the population responses for different stimuli are uncorrelated, whereas with pseudosparseness the

population responses are correlated. Pseudosparseness is defined as the mean correlation between population responses to different stimuli.

We suggest that the difference between pseudosparseness and authentic sparseness may be understood when considering the degree of statistical heterogeneity amongst responses across a population. With authentic sparseness, the average response across the stimulus set for each neuron is roughly the same across the population. This can be seen in mean responses for different neurons in the example shown in Fig. 1b, where the curve is flat. With authentic sparseness, the response heterogeneity across the population is low. On the other hand, when pseudosparseness is large, individual neurons tend to have responses that are either consistently high or consistently low for all stimuli, relative to those of other neurons. This can be seen in responses in the example shown in Fig. 1e. With high pseudosparseness, in other words, the response heterogeneity across the population is high.

In 12 neurophysiological data sets covering different visually responsive cortical areas in macaque monkeys, we found consistently high levels of pseudosparseness, which we would attribute to heterogeneous responses across the population. In most of these data sets, a high sparseness measure is accompanied by a high pseudosparseness measure, leading to the possibility that what is being reported as high sparseness may actually be pseudosparseness.

In some cases within the literature, responses of a neuron are normalized by dividing them by their mean response or alternatively subtracting the mean (and possibly dividing by the standard deviation). While this normalization can be useful for identifying which are the most effective stimuli for individual neurons, having a different normalization factor for each neuron disrupts population correlations leading to pseudosparseness values artifactually close to zero.

High pseudosparseness reduces encoded information. Higher pseudosparseness means that a neural population is transmitting less Shannon information about different stimuli (holding other aspects constant when calculating information, such as the stimulus set statistics, discretization of the neural responses, and so forth as discussed by Richmond and Optican[45]). A consequence of the high pseudosparseness observed in visual cortical data is that neural populations are encoding substantially less information than would be expected under sparse coding. By definition, high pseudosparseness means high correlation of responses across the population for different stimuli. When responses are highly correlated, information decreases. For example, if the population responses for all stimuli are perfectly correlated (as in Fig. 1d–f, pseudosparseness = 1.0), then the population responses are identical regardless of the stimuli, and Shannon information is zero.

Interestingly, the values of AIT and LIP pseudosparseness indices reverse for shape and space. For shape stimuli, the pseudosparseness index is lower in AIT than LIP (Fig. 6a, b). For location stimuli, the pseudosparseness index is lower in LIP than AIT (Fig. 6d, e). As indicated by the pseudosparseness index, AIT transmits more information about shape stimuli than LIP, while LIP transmits more location information than AIT.

We created a neural model to examine which parameters are important for generating pseudosparseness. It consists of a mosaic of neurons with Gaussian receptive fields (with the receptive fields located in a feature space and not necessarily in physical location space) and with a random additive response offset (spontaneous or baseline activity) for each neuron (Fig. 7, first row). In our model, the random response offsets are important for creating response heterogeneity across the neural population, leading to high pseudosparseness. Another factor

affecting response heterogeneity and leading to high pseudosparseness is a stimulus set that poorly covers portions of the feature space defined by the encoding neural population (Fig. 7, second row relative to third row). When some areas of the feature space are well covered and others are not (small stimulus set or high receptive field dispersion), that increases heterogeneity in responses across a population as a whole.

In conclusion, given the observation of consistently high values of pseudosparseness in the neurophysiological data, prior reports of sparse coding need be reconsidered, and serious reflection should be given to the degree to which authentic sparseness provides a useful framework for understanding neural coding in cortex.

## Methods

**Monkey visual cortex data sets.** We reanalyzed a number of previously published neurophysiological data sets from different regions of visually responsive cortex in macaque monkeys. Animals' care was in accordance with institutional guidelines, as indicated in the references for the various data sets. Data from V1 and V2 were collected from anesthetized preparations, while other data sets used awake, behaving animals. All data were recorded extracellularly. Single-electrode data were sorted to identify single neurons, while multielectrode data (both Utah array and tetrode) consisted of both well-isolated single units and small clusters of multiunit activity. Each response spectrum was almost always determined from data from a single monkey. In a few cases, data were pooled from two monkeys to form a single spectrum, as noted in the "Results" sections. Collectively, we present data from 11 monkeys.

Characteristics of the data sets are summarized below for each cortical area. Further details can be found in their respected published references.

*V1 a.* These V1 data (plotted in Fig. 2a–d) were recorded using a Utah multi-electrode array with a 10 × 10 array of microelectrodes and 400 μm spacing. Cells were recorded during ten recording sessions taken from three anesthetized monkeys. The population sizes in the sessions were N = [26 37 34 28 44 16 36 66 29 76] neurons. For data analysis, we selected two sessions from two monkeys, session 5 (N = 44) and session 8 (N = 66). Units consist of well-isolated single neurons, as well as small multiunit clusters. The stimulus set consisted of 416 gratings and 540 natural images that were flashed during presentation.

To compare multielectrode and single-electrode population statistics (sparseness and pseudosparseness), single-electrode population data were created by selecting one neuron randomly from each recording session to create a sequential population with N = 10 neurons, as described in "Comparing population statistics using single-electrode and multielectrode data" in the "Methods" section and "Pseudosparseness in the monkey visual cortex" in the "Results" section.

These data were collected in the laboratory of Adam Kohn at the Albert Einstein College of Medicine and downloaded from http://crcns.org/data-sets/vc/pvc-8 (ref. [30]). They were originally published by Coen-Cagli, Kohn, and Schwartz[29].

*V1 b.* The data (plotted in Fig. 3c) consists of N = 24 neurons recorded with single electrodes from an anesthetized monkey. The cells were stimulated by a set of 157 synthetic stimulus patterns consisting of 78 random textures and 79 shaded paraboloid figures. These V1 cells were complex cells. They were originally published by Lehky, Sejnowski, and Desimone[14,31].

*V2.* These V2 multielectrode data (plotted in Fig. 5a) consists of a session with N = 37 neurons recorded using a set of tetrodes from an anesthetized monkey. Units consisted of both well-isolated single units and multiunit activity. The stimuli were oriented gratings.

The data were collected in the laboratory of Adam Kohn at the Albert Einstein College of Medicine and downloaded from http://crcns.org/data-sets/vc/v1v2-1 (ref. [34]). They were originally published by Zandvakili and Kohn[46], as well as Semedo et al.[33].

*MT.* These data (plotted in Fig. 5b) contain N = 45 neurons collected from two monkeys using single electrodes. The stimuli were naturalistic videos, which contained full-screen natural movies that were motion enhanced with an overlay of textured three-dimensional moving objects. Simulated saccadic eye movements were introduced by cutting the movies into segments and shuffling their order, similar to the sparse coding study of Vinje and Gallant[10]. This manipulation was intended to mimic free viewing of stimuli. The videos were projected at 83 Hz (12.0 ms per frame).

Out of the video frames, 200 snippets were used to create 200 stimuli for each cell. Each snippet was spaced 100 frames (1.2 s) along the video and had a duration

of ten frames (120 ms). Spike count over the 120 ms stimulus duration defined the stimulus response.

The data were collected in the laboratory of Jack Gallant at the University of California Berkeley and downloaded from http://crcns.org/data-sets/vc/mt-2 (ref. [35]). The data were originally published in Nishimoto and Gallant[36].

*Perirhinal.* These data (plotted in Fig. 5c) contains $N = 92$ neurons collected using single electrodes. The stimuli consisted of 110 object images, both natural and man-made objects, against a neutral background. The data were originally published by Lehky and Tanaka[37].

*Anterior inferotemporal cortex a.* These single-electrode recordings (plotted in Fig. 5d) contain data for $N = 122$ neurons. This stimulus set was identical to that used in the Prh data (data set 5), with 110 object images. The data were originally published by Lehky and Tanaka[37].

*Anterior inferotemporal cortex b (shape).* The data here (plotted in Fig. 6a) consists of $N = 85$ neurons obtained through single-electrode recording. The stimulus set had eight shapes that were simple geometric shapes. The data were originally published by Lehky and Sereno[12].

*Anterior inferotemporal cortex c (location).* These data (plotted in Fig. 6d) consists of $N = 83$ neurons using single-electrode recording. The stimulus set consisted of eight retinotopic locations across the visual field, allowing construction of a population response to different locations. The shape stimulus was the most responsive shape for each neuron in the population. The data were originally published by Lehky et al.[40] and Sereno and Lehky[11].

*Lateral intraparietal cortex a (shape).* This single-electrode data set (plotted in Fig. 6b) contains $N = 53$ neurons. The stimulus set consisted of eight shapes that were simple geometric shapes, the same shapes used in AIT (data set 7). The data were originally published by Sereno and Amador[38] and Lehky and Sereno[12].

*Lateral intraparietal cortex b (location).* This data set (plotted in Fig. 6e) consists of $N = 83$ neurons obtained through single-electrode recording. The stimulus set consisted of eight retinotopic locations across the visual field, allowing construction of a population response to different locations. The shape stimulus was the most responsive shape for each neuron in the population. The data were originally published by Sereno and Amador[38] and Sereno and Lehky[11], and follows the same procedures as used in AIT (data set 8).

*Frontal eye field a (shape).* This data set (plotted in Fig. 6c) consists of $N = 72$ neurons obtained through single-electrode recording. The stimulus set had eight shapes that were simple geometric patterns, the same stimuli previously used in AIT and LIP by Sereno and Amador[38], and Lehky and Sereno[12] (data sets 7 and 9). The data were originally published by Peng, Sereno, Silva, Lehky, and Sereno[39].

*Frontal eye field b (location).* These data (plotted in Fig. 6f) are an unpublished part of the Peng et al.[39] data set, now dealing with stimulus location rather than stimulus shape. It comprises 62 neurons stimulated by a set of eight location stimuli (stimuli placed at different locations of the visual field), collected using single-electrode recording. The procedures for these FEF location data are analogous to those for the previous AIT location data (Lehky et al.[40], and Sereno and Lehky[11]) and LIP location data (Sereno and Amador[38], and Sereno and Lehky[11]; data sets 8 and 10).

**Pseudosparseness index.** Given a neural population, each stimulus produces a particular population response vector. The pseudosparseness index is defined as the mean Pearson correlation between the population responses for all pairs of stimuli. The mean correlation is determined by transforming each correlation coefficient $r$ using Fisher's $z$ transform, $z = \operatorname{atanh}(r)$, calculating the mean of the $z$ values, and then using the inverse transform $\bar{r} = \tanh(\bar{z})$[47]. Fisher's $z$ transform serves to normalize the sampling of correlation coefficients, making the estimate of mean correlation less affected by distribution skew.

**Comparing population statistics using single-electrode and multielectrode data.** We had V1 data available from a Utah multielectrode array, described in "Monkey visual cortex data sets" of the "Methods" section, data set 1. These data included ten recording sessions using the array, with all sessions using identical recording methods and the same stimulus set.

We wished to compare multielectrode and single-electrode population statistics (both sparseness and pseudosparseness) from these data. For the multielectrode data set, we selected one recording session (session 8) with $N = 66$ neurons. Single-electrode population data were created by selecting one random neuron from each of the ten recording sessions to create a single sequential population with $N = 10$ neurons. The $N = 10$ single-electrode population was repeated 10,000 times with random selections of neurons across the recording sessions for each repetition.

Because population statistics might be sensitive to population size, we randomly subsampled the $N = 66$ neurons from the multielectrode set into subpopulations of

sizes $N = [5\ 10\ 20\ 30\ 40\ 50\ 60]$. Neurons in subpopulations were selected through random permutation (Matlab command `randperm(N,k)`, with $N = 66$ and $k$ = subpopulation size). Each subpopulation size was repeated 10,000 times. We generated the entire curves of population statistics as a function of population size for completeness, although the critical comparison here is that between the $N = 10$ single-electrode population and the $N = 10$ multielectrode population.

**Statistics and reproducibility.** This study involves a reanalysis of previously published data collected by different investigators from different visually responsive cortical areas of macaque monkeys. A total of 11 monkeys were used, with data from 1 or 2 monkeys for each cortical data set. Although details differ for each data set, each contained a sample of several dozen neurons, with responses for each neuron repeated at least five times and more typically ten times. The data analysis centered around two parameters, sparseness and pseudosparseness, which are explained in the "Introduction" and "Methods" sections. These parameters were quantified using standard measures, such as mean, median, and the Pearson correlation coefficient. Variation is generally given as standard deviation although standard error is occasionally used, as indicated in the text. Two-sided values of significance are given, and confidence levels are at the 95% level. Statistical powers of comparisons are provided. In some cases, bootstrap resampling was used to determine statistics, as described in the appropriate sections.

**Reporting summary.** Further information on research design is available in the Nature Research Reporting Summary linked to this article.

## Data availability

Data associated with Figs. 2, 3a, 4, and 5a, b are available from crcns.org (Collaborative Research in Computational Neuroscience), as detailed in the "Monkey visual cortex data set" in the "Methods" section. Data associated with Figs. 3b, 5c, d, and 6 are available via the corresponding author upon reasonable request.

## Code availability

Matlab code for the pseudosparseness function can be downloaded at https://github.com/slehky/pseudosparseness-commsbio.

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

## Acknowledgements
This study was partially supported by start-up funds from Purdue University. The study included analysis of data from the Collaborative Research in Computational Neuroscience (CRCNS) program (crcns.org).

## Author contributions
S.R.L. designed the study, provided data, carried out data analysis and modeling, interpreted the findings, and wrote and revised the manuscript. A.B.S. provided data, interpreted the findings, and wrote and revised the manuscript. K.T. provided data, interpreted the results, and revised the manuscript.

## Competing interests
The authors declare no competing interests.
