## [Peer Review File · Communications Biology]

Reviewers' comments:

Reviewer #1 (Remarks to the Author):

The paper addresses an interesting question -- whether measurements of sparseness in the brain reflect genuinely sparse coding, or whether these measurements are exaggerated by correlations between neurons. The authors refer to this exaggeration as pseudosparseness. They measure pseudosparseness in data from previous experiments, and show that it is high. This indicates that measurements of sparseness may be overestimates.

The paper addresses a genuine, neglected issue, and is clearly written for the most part. However, I think there are two major problems with the presentation. First, the experiments are not well designed for accurate estimation of pseudosparseness -- they use non-naturalistic stimulation protocols, and the neuronal responses are recorded sequentially. Both of these are likely to affect measured pseudosparseness values.

Second, the paper would benefit from more thorough investigation and description in several areas. Given that the data come from previous experiments, I think it would be desirable to use a much wider range of data from different species and parts of the brain. Also, no mathematical investigation is given into the relationship between pseudosparseness and sparseness measurements. Finally, few details are given about the experimental protocols used. If these areas were addressed, I believe the paper would be much more convincing.

1. All experiments use non-naturalistic stimulus protocols. Proponents of sparse coding claim that neurons represent natural stimuli with a sparse code. This does not necessarily imply that neurons represent all stimuli with a sparse code. To be sure that the pseudosparseness measurements are actually relevant to this claim, it would be necessary to use natural stimuli in the experiments. Instead, however, the temporal structure of the protocols are non-natural in all cases, and only one experiment used stimuli with naturalistic spatial structure (as far as I can tell from the limited information given). It is possible that the high degree of correlation between neurons (pseudosparseness) arises from the structure of these protocols -- for example, neurons may be activated together if stimuli flash on at a particular time. Such correlations may not be present during naturalistic stimulation (e.g. freeviewing). I think that pseudosparseness measurements during naturalistic stimulation are essential, if the paper is to provide convincing evidence that pseudosparseness is not just an artefact of particular stimulation protocols.

2. Use of sequential recordings. In all cases, the recordings were made using single electrodes. Thus, the recordings must have been made sequentially, and the "population response" was constructed afterwards by treating these recordings as simultaneous. This approach is problematic because it may not correctly estimate the level of correlation between neurons in a simultaneously recorded population. Any correlations that arise from shared neuronal inputs will be missed in this analysis. I think the paper would be significantly improved by including a simultaneously recorded data set.

3. Breadth of experiments. All of the experiments used macaques, and are from the areas associated with vision. As a result, it is unclear whether pseudosparseness is restricted to macaque vision, or whether it is a more widespread phenomenon. The generality of the results would be increased by including data from a much wider range of brain areas and species. Since the data were gathered from the literature, it should be possible to obtain data from other labs for this.

4. Definition of pseudosparseness and relationship with true sparseness. No justification is given for the definition of pseudosparseness. The effects of pseudosparseness on sparseness measurements are not explained in any detail. Ideally, the paper would provide a mathematical relationship which quantifies the relationship between measured sparseness, true sparseness and pseudosparseness. Less ideally, the effects of pseudosparseness on measured sparseness could be

quantified for a wide range of reasonable values (the presented model seems appropriate for this). In either case, the effect of pseudosparseness on sparseness measurements needs to be demonstrated and explained quantitatively.

pg. 11 "A higher pseudosparseness index means that a neural population is transmitting less Shannon information about different stimuli. This follows from the fact that higher pseudosparseness indicates that responses across the population are more correlated across the stimulus set." This is an interesting statement, and may well be true. However, it is not obviously correct because Shannon information reflects more than just correlation. Some support for this statement is needed.

5. Experimental details. Limited details are given about the experiments. To a degree, it is possible to work out the details by referring back to the previous papers. But the information given in the present manuscript is not sufficient to evaluate the validity of the pseudosparseness measurements. Helpful information would include the number of animals used, the details of spike sorting, the response rates of the neurons involved.

Minor points

pg. 14 "a anaesthetized" should read "an anaesthetized"

pg. 17 "result we are interested it" should read "result we are interested in"

Reviewer #2 (Remarks to the Author):

The paper "Pseudosparseness in the visual system" is a theoretical study of population coding of visual stimuli by populations of neurons in various areas of the visual cortex. I think this is a brilliant paper written with great insight. The insights in the paper have very important consequences for data analysis of neuronal populations.

That said, the paper could be improved so that the large target audience can understand it. The arrangement of the paper—perhaps necessitated by the journal's word count limit?—is part of the problem of lack of clarity. References in Results to equations that appear only later in Materials and Methods are a stumbling block. Also placement of the definition of population sparseness in the Methods makes the text in the Introduction and Results unclear to non-experts. I suggest re-arranging the presentation so that crucial definitions and equations are presented before they are used in the exposition. Also, I think something else is missing that would help the reader understand the paper better, and that is an explanation in this paper for why population sparseness was defined as it was in the literature. Without understanding the reason for the definition, it isn't clear to the reader why such a misleading definition was used in the first place.

Finally, I have a question about the model for pseudosparseness. The authors tell us that the main characteristic of neurons that causes pseudosparseness in their model is response offset --this is what makes some neurons respond strongly to many stimuli and others respond very little to many stimuli, causing pseudosparseness. Perhaps another factor is at work in the real cortex, and I wonder whether or not the authors have considered it or could consider it as a more plausible explanation for pseudosparseness. That factor is variation in response gain or sensitivity across the population. It seems more natural to suppose that different neurons would have more gain than others caused by different patterns of functional connectivity. If the authors have considered heterogeneity in gain and it did not work, that would be useful to know. But if they haven't considered it, adding such an analysis might be useful in persuading neuroscientists.

Responses to reviewers' comments:

Reviewer #1 (Remarks to the Author):

The paper addresses an interesting question -- whether measurements of sparseness in the brain reflect genuinely sparse coding, or whether these measurements are exaggerated by correlations between neurons. The authors refer to this exaggeration as pseudosparseness. They measure pseudosparseness in data from previous experiments, and show that it is high. This indicates that measurements of sparseness may be overestimates.

The paper addresses a genuine, neglected issue, and is clearly written for the most part. However, I think there are two major problems with the presentation. First, the experiments are not well designed for accurate estimation of pseudosparseness -- they use non-naturalistic stimulation protocols, and the neuronal responses are recorded sequentially. Both of these are likely to affect measured pseudosparseness values.

Second, the paper would benefit from more thorough investigation and description in several areas. Given that the data come from previous experiments, I think it would be desirable to use a much wider range of data from different species and parts of the brain. Also, no mathematical investigation is given into the relationship between pseudosparseness and sparseness measurements. Finally, few details are given about the experimental protocols used. If these areas were addressed, I believe the paper would be much more convincing.

1. All experiments use non-naturalistic stimulus protocols. Proponents of sparse coding claim that neurons represent natural stimuli with a sparse code. This does not necessarily imply that neurons represent all stimuli with a sparse code. To be sure that the pseudosparseness measurements are actually relevant to this claim, it would be necessary to use natural stimuli in the experiments. Instead, however, the temporal structure of the protocols are non-natural in all cases, and only one experiment used stimuli with naturalistic spatial structure (as far as I can tell from the limited information given). It is possible that the high degree of correlation between neurons (pseudosparseness) arises from the structure of these protocols -- for example, neurons may be activated together if stimuli flash on at a particular time. Such correlations may not be present during naturalistic stimulation (e.g. freeviewing). I think that pseudosparseness measurements during naturalistic stimulation are essential, if the paper is to provide convincing evidence that pseudosparseness is not just an artifact of particular stimulation protocols.

We have added data for flashed natural image stimuli for V1 in Figure 2ab. We have also added data for naturalistic video stimuli for area MT from Jack Gallant's data in Figure 5b. The MT stimuli include simulated saccades that

mimic free viewing, as described by Nishimoto and Gallant (2011) following the methodology of Vinje and Gallant (2000).

2. Use of sequential recordings. In all cases, the recordings were made using single electrodes. Thus, the recordings must have been made sequentially, and the "population response" was constructed afterwards by treating these recordings as simultaneous. This approach is problematic because it may not correctly estimate the level of correlation between neurons in a simultaneously recorded population. Any correlations that arise from shared neuronal inputs will be missed in this analysis. I think the paper would be significantly improved by including a simultaneously recorded data set.

We have added multielectrode data for V1 in Figure 2 and for V2 in Figure 5a. In addition, we have added a comparison between multielectrode and single-electrode data in Figure 4.

3. Breadth of experiments. All of the experiments used macaques, and are from the areas associated with vision. As a result, it is unclear whether pseudosparseness is restricted to macaque vision, or whether it is a more widespread phenomenon. The generality of the results would be increased by including data from a much wider range of brain areas and species. Since the data were gathered from the literature, it should be possible to obtain data from other labs for this.

We have added data from three more areas of macaque cortex, namely V2, MT, and perirhinal cortex. This increases the number of cortical areas from 4 to 7 (V1, V2, MT, AIT, LIP, FEF, and Prh) and includes areas from all four lobes of primate cortex (occipital, temporal, parietal, and frontal). We have not extended the data from monkeys to rats and other species, nor did we extend the data from vision to audition or other sensory modalities. These requests go beyond what we perceive to be the scope of the work. Our interest and expertise is in primate vision and we believe that the paper covers an important and sufficiently broad topic for an initial and impactful presentation of the concepts discussed in the manuscript.

4. Definition of pseudosparseness and relationship with true sparseness. No justification is given for the definition of pseudosparseness. The effects of pseudosparseness on sparseness measurements are not explained in any detail. Ideally, the paper would provide a mathematical relationship which quantifies the relationship between measured sparseness, true sparseness and pseudosparseness. Less ideally, the effects of pseudosparseness on measured

sparseness could be quantified for a wide range of reasonable values (the presented model seems appropriate for this). In either case, the effect of pseudosparseness on sparseness measurements needs to be demonstrated and explained quantitatively.

This is an interesting question that remains beyond the scope of this paper. There is no standard neural model of sparseness (as opposed to simply calculating sparseness of data). A variety of statistical models can lead to identical sparseness, yet perhaps produce different pseudosparseness values. How the parameters for possible sparseness models interact with those for pseudosparseness is complex and needs to be explored as a project itself and not as an addendum to the current one.

pg. 11 "A higher pseudosparseness index means that a neural population is transmitting less Shannon information about different stimuli. This follows from the fact that higher pseudosparseness indicates that responses across the population are more correlated across the stimulus set." This is an interesting statement, and may well be true. However, it is not obviously correct because Shannon information reflects more than just correlation. Some support for this statement is needed.

We have clarified the statement as follows:

Higher pseudosparseness means that a neural population is transmitting less Shannon information about different stimuli (holding other aspects constant when calculating information, such as the stimulus set statistics, discretization of the neural responses, and so forth as discussed by Richmond and Optican (1987)). A consequence of the high pseudosparseness observed in visual cortical data is that neural populations are encoding substantially less information than would be expected under sparse coding. By definition, high pseudosparseness means high correlation of responses across the population for different stimuli. When responses are highly correlated, information decreases. For example, if the population responses for all stimuli are perfectly correlated (as in Figure 1b, pseudosparseness=1.0), then the population responses are identical regardless of the stimuli, and Shannon information is zero.

5. Experimental details. Limited details are given about the experiments. To a degree, it is possible to work out the details by referring back to the previous papers. But the information given in the present manuscript is not sufficient to evaluate the validity of the pseudosparseness measurements. Helpful information would include the number of animals used, the details of spike sorting, the response rates of the neurons involved.

The mean and standard deviation of the response for every individual unit is given by the y-axis of all the spectrum plots (Figures 2, 3, 5, 6). We have added the number animals used in the text of the manuscript. We have also added basic information about spike sorting. Single electrode data were sorted to identify single neurons, while multielectrode data consisted of both well-isolated single units and small clusters of multiunit activity. Further details about spike sorting are available in the published references.

Minor points

pg. 14 "a anaesthetized" should read "an anaesthetized"

pg. 17 "result we are interested it" should read "result we are interested in"

Corrected

Reviewer #2 (Remarks to the Author):

The paper "Pseudosparseness in the visual system" is a theoretical study of population coding of visual stimuli by populations of neurons in various areas of the visual cortex. I think this is a brilliant paper written with great insight. The insights in the paper have very important consequences for data analysis of neuronal populations.

1. That said, the paper could be improved so that the large target audience can understand it. The arrangement of the paper—perhaps necessitated by the journal's word count limit?—is part of the problem of lack of clarity. References in Results to equations that appear only later in Materials and Methods are a stumbling block. Also placement of the definition of population sparseness in the Methods makes the text in the Introduction and Results unclear to non-experts. I suggest re-arranging the presentation so that crucial definitions and equations are presented before they are used in the exposition. Also, I think something else is missing that would help the reader understand the paper better, and that is an explanation in this paper for why population sparseness was defined as it was in the literature. Without understanding the reason for the definition, it isn't clear to the reader why such a misleading definition was used in the first place.

We have added a paragraph in the Introduction section (second paragraph in the section) providing more background about sparseness, including a definition of sparseness and references. Also, we have moved the equation for sparseness from the Methods section to the Introduction. Finally, more information about the pseudosparseness model has been moved from the Methods section to the main body of the manuscript, including the central equation for the model.

2. Finally, I have a question about the model for pseudosparseness. The authors tell us that the main characteristic of neurons that causes pseudosparseness in their model is response offset –this is what makes some neurons respond strongly to many stimuli and others respond very little to many stimuli, causing pseudosparseness. Perhaps another factor is at work in the real cortex, and I wonder whether or not the authors have considered it or could consider it as a more plausible explanation for pseudosparseness. That factor is variation in response gain or sensitivity across the population. It seems more natural to suppose that different neurons would have more gain than others caused by different patterns of functional connectivity. If the authors have considered heterogeneity in gain and it did not work, that would be useful to know. But if they haven't considered it, adding such an analysis might be useful in persuading neuroscientists.

We have added a Gaussian-distributed gain factor to Eq. 2. Gain is now discussed on page 14. Increasing the mean of the gain causes pseudosparseness to decrease while changing the standard deviation of the gain has little effect.

References

- Nishimoto, S., & Gallant, J. L. (2011). A three-dimensional spatiotemporal receptive field model explains responses of area MT neurons to naturalistic movies. *Journal of Neuroscience*, 31, 14551-145564.
- Richmond, B. J., & Optican, L. M. (1987). Temporal encoding of two-dimensional patterns by single units in primate inferior temporal cortex. III. Information theoretic analysis. *Journal of Neurophysiology*, 57, 162-178.
- Vinje, W. E., & Gallant, J. L. (2000). Sparse coding and decorrelation in primary visual cortex during natural vision. *Science*, 287, 1273-1276.

REVIEWERS' COMMENTS:

Reviewer #2 (Remarks to the Author):

In the revised paper, the authors have responded to and answered my comments on the original submission. I was very positive about the insights in this paper when it was originally submitted, and it has become a stronger paper upon revision.

Reviewer #3 (Remarks to the Author):

This manuscript investigates the possibility that previous estimates of sparse coding have been exaggerated due to the presence of significant correlations between cell activation vectors, at least in the visual system. These correlations give rise to what the authors called "pseudosparseness". It is clearly in the interest of the field to distinguish between these two possibilities, as they have potentially deep implications to understand neural coding.

I think that this revised version of the manuscript does a wonderful job of introducing the reader gently to the problem at hand. If anything, I had the slight impression that the authors could have mounted an attack towards a straw man, in the sense that I'm not fully confident previous studies have neglected the presence of positive significant correlations to the extent claimed by the authors. However, I am not that familiar with the literature on this topic.

My main question is whether the correlations could arise due to composite stimuli being coded as a response across overlapping groups of cells. If the neurons are responding to certain primitive features of the stimuli A, B, C, and features B and C are present through several stimuli but A is being replaced, then in principle I do not find strange that correlations exist between response vectors. Adopting a unique response vector per stimulus appears to be a highly inefficient way to code for this information, and in this sense it didn't come as a surprise that the coding was "pseudosparseness" to me.

I am inclined to believe this situation could be especially problematic for natural stimuli. However the authors found similar pseudosparseness for natural stimuli and gratings. I'm not sure I understand why this happened, perhaps the authors could comment on this further?

With regards to the authors' responses to the concerns raised by reviewer #1:

Yes, the authors addressed the concerns raised by reviewer #1. They added new data comprising naturalistic stimuli as requested and they also expanded the recording modalities as requested, in particular, to include non sequential recordings (multielectrode arrays). The only request they didn't fulfill was to incorporate data from other species. However I agree with the authors that this is not strictly necessary and it is also tricky. Ideally they should have included data from a species with a similarly developed visual system so rats are not ideal. Humans are the obvious choice to compare with nonhuman primates but these are very invasive recordings so the data is almost nonexistent for humans. Also, in principle I don't see why these results would hold for nonhuman primates and not for other related species.

Response to Reviewer 3

Reviewer #3 (Remarks to the Author):

This manuscript investigates the possibility that previous estimates of sparse coding have been exaggerated due to the presence of significant correlations between cell activation vectors, at least in the visual system. These correlations give rise to what the authors called "pseudosparseness". It is clearly in the interest of the field to distinguish between these two possibilities, as they have potentially deep implications to understand neural coding.

I think that this revised version of the manuscript does a wonderful job of introducing the reader gently to the problem at hand. If anything, I had the slight impression that the authors could have mounted an attack towards a straw man, in the sense that I'm not fully confident previous studies have neglected the presence of positive significant correlations to the extent claimed by the authors. However, I am not that familiar with the literature on this topic.

1. My main question is whether the correlations could arise due to composite stimuli being coded as a response across overlapping groups of cells. If the neurons are responding to certain primitive features of the stimuli A, B, C, and features B and C are present through several stimuli but A is being replaced, then in principle I do not find strange that correlations exist between response vectors. Adopting a unique response vector per stimulus appears to be a highly inefficient way to code for this information, and in this sense it didn't come as a surprise that the coding was "pseudosparse" to me.

I am inclined to believe this situation could be especially problematic for natural stimuli. However the authors found similar pseudosparseness for natural stimuli and gratings. I'm not sure I understand why this happened, perhaps the authors could comment on this further?

(Note: Reviewer 1 never responded to the extensive changes they requested for the revised manuscript, and thus Reviewer 3 was brought in.)

Response: Using the same V1 neural population, we show that the response spectra to synthetic stimuli (e.g. gratings) and natural stimuli are almost identical (Figure 2). The close similarity of responses raises the possibility that pseudosparseness may reflect some low-level biophysical characteristic across the population. In particular, response offset (essentially spontaneous or baseline activity), which is an important factor in the pseudosparseness model, may be a fixed biophysical characteristic for individual neurons in the population (although variable for different neurons across the population), thus leading to similar profiles for different stimulus sets. We have added new data analyses in the Results section (bottom of page 7) as well as briefly discuss the relationship

between pseudosparseness for grating and natural stimuli in the Discussion section (page 18).

With regards to the authors' responses to the concerns raised by reviewer #1:

Yes, the authors addressed the concerns raised by reviewer #1. They added new data comprising naturalistic stimuli as requested and they also expanded the recording modalities as requested, in particular, to include non sequential recordings (multielectrode arrays). The only request they didn't fulfill was to incorporate data from other species. However I agree with the authors that this is not strictly necessary and it is also tricky. Ideally they should have included data from a species with a similarly developed visual system so rats are not ideal. Humans are the obvious choice to compare with nonhuman primates but these are very invasive recordings so the data is almost nonexistent for humans. Also, in principle I don't see why these results would hold for nonhuman primates and not for other related species.